# "O Piteous Spectacle! O Bloody Times!": The Faithlessness of English Identity in *1, 2,* and *3 Henry VI*

Matthew Carter

Department of English, Clayton State University, Morrow, GA 30260, USA; matthewcarter2@clayton.edu

**Abstract:** Shakespeare's Henry VI trilogy is jam-packed with spectacle: heads are severed and made to kiss, women dress as men and lead armies, ghosts predict the future, and a plethora of miracles take place all over the various locales we visit across fifteen acts. In fact, if "faith is... the evidence of things not seen," as asserted by the author of Hebrews, then the Henry VI plays are entirely devoid of faith, by the merit of bringing miraculous events from the realm of faith into the realm of observational knowledge. Of note, then, is the fact that the trilogy depicts Henry as a weak king whose main virtue is his commitment to his faith. Compared to other kings in Shakespearean history plays, Henry is almost-constantly referencing the spiritual world, and the world he lives in is so full of miraculous happenings that miracles themselves run the risk of becoming banal. Perhaps surprisingly, given the trilogy's thematic investment in miracles and spirituality, the English are defined in the plays as destroying or debunking miraculous spectacles. From Gloucester outsmarting Simpcox in his feigned healing to the putting-down of two witches (Joan in 1 Henry VI and Margery Jourdain in 2 Henry VI), it seems that, despite Henry's incredible devotion, his courtiers raise skepticism to the level of modus operandi. In this essay, I hope to examine the way that the second Henriad depicts a version of England that places logic and skepticism in the seat of faith, while its ruler's faith is often both uninterrogated and misplaced. Shakespeare stages a teleology of spectacle that highlights English faithlessness as a source of internecine struggle and insurrection, while also cautioning against naivete in the face of canny nemeses.

**Keywords:** faith; doubt; atheism; Christianity; early modern; Shakespeare

In early modern England, observational knowledge was treated as a pathway to understanding religious truth, but also as a panacea against questionable forms of "superstition". The Protestant drive to dispel Catholic frivolities led many in the country to seek a balance between unproveable faith and observational data. Francis Bacon, for instance, argues that "There is a superstition in avoiding superstition, when men think to do best if they go furthest from the superstition formerly received: therefore care would be had that (as it faireth in ill purgings) the good be not taken away with the bad, which commonly is done, when the people is the reformer" ([Bacon 1985](#), p. 112). To put it simply, faith was a double-edged sword, capable of damaging the commonweal and establishing what Bacon calls "an absolute monarchy in the minds of men" (ibid., p. 111), despite the fact that Bacon's opinion of then-modern skeptics was at-best unflattering: "there be not so much blood in them as was in those of the ancients" (ibid., p. 61).

This disconnect between faithfulness and skepticism is perhaps best exemplified in Shakespeare's *1, 2,* and *3 Henry VI*, plays which are heavily invested in questions of faith, miracles, and skepticism of both.[1] Henry VI himself is characterized as a devotedly faithful man, while his subjects operate in a consistently skeptical mode of thinking. By characterizing the English as skeptical and doubtful, Shakespeare positions the men and women who lost Henry V's conquered French territories as religious nonconformists whose doubt directly brings about their downfall. Shakespeare stages a teleology of spectacle that highlights English faithlessness as a source of internecine struggle and insurrection,

while also cautioning against naivete in the face of canny nemeses. Frequently, scholars have focused on the more-than-usual religiosity of the *Henry VI* trilogy as a focus that Shakespeare would not maintain in later plays. John A. Warrick offers the unique insight that Shakespeare seems to have drawn directly from the hyper-religious medieval dramas of the past, deploying "a dramaturgical model that substantially draws upon conventions unique to medieval religious theatre" (Warrick 2016, p. 58). Jean-Christophe Mayer takes it even further, exerting that "faith in Shakespeare is not so much a matter of systematic allegiance as one of constant debate and controversy" (Mayer 2013, p. 74), and specifically claims that Shakespeare's representation of faith in the trilogy highlights an "implicit cynicism on matters of religion", which he later modifies with *King John* (Mayer 2013, p. 75). However, the plays' treatment of religion might be more complicated than even Mayer's nuanced reading of it. In this study, I shall use the Geneva Bible's translation of Hebrews 11:1 because of its common presence in early modern English devotional literature, and so that I might deploy its working definition of faith: "Now faith is the grounds of things which are hoped for, and the evidence of things which are not seen". The author of Hebrews characterizes faith as that which provides evidence for something that yields no tangible evidence in and of itself. Miracles, then, provide spectacular breakages of the laws of reality—resurrection is impossible, for instance, but faith provides context for the miraculous and verifies what cannot be otherwise witnessed. Because Henry is characterized as faithful to a fault, it would make sense that his faithfulness might be perceived as a weakness in his leadership qualities, but as the verse describes, faith is a supplement to knowledge, not simply a replacement for it. Faith can fill in the gaps in one's knowledge, but observation, especially the observation of the spectacular, can both increase knowledge and reafirm one's faith. Understanding the interplay between spectacle, which leads to knowledge, and faith, which exists parallel to the evidentiary world, highlights some of the tensions Shakespeare explores in the *Henry VI* plays. To understand the complex negotiation of faith and knowledge that Shakespeare participates in in the *Henry VI* plays, we can see how moments of faith are disrupted. The *Henry* plays provide meaningful interrogations of miraculous events in the figures of Simon Simpcox and Joan of Arc, while also using the persistent theme of witchcraft as a compass for navigating the fraught waters of faith. Because the plays are heavily invested in religion and in religious thinking, but often undercut the miraculous events that would solidify faith, Shakespeare invites his audience to consider the role of skepticism and investigation in Christian praxis, offering negative examples of the spectacular and the miraculous, while demonstrating the strengths of healthy skepticism. The skepticism towards miracles in turn serves as a microcosm for other instantiations of faith, such as Henry's court failing to have faith in his leadership, or the French court placing too much faith in Joan's witchcraft.

One of the most prominent examples of English skepticism toward miracles in the tetralogy circles around the figure of Simpcox. Simpcox claims to be "Born blind, an't please your grace" (2.2.1.76)[2]. Henry's arrival in town corresponding with Simpcox's receiving of his sight is interpreted as a miracle, and Henry's response to the news of Simpcox's sightedness is telling: "Now God be praised, that to believing souls,/Gives light in darkness, comfort in despair" (2.2.1.65-66). The response is unclear as to whether Henry believes that Simpcox has received his sight because Simpcox was visiting the shrine at St. Alban's, or if Henry's presence in town has caused this miracle to happen. This vague statement about "believing souls" cuts two, if not three, ways, as it suggests that belief has the power to make miracles happen and that Simpcox is evidence of this teleological fact. However, as we find out, Simpcox's miracle is a false one, and Henry becomes the "believing soul"—one who comes off as naive, rather than pious. The ability to interpret the difference between fake and real impairment is referenced heavily in the period—fear that "beggars who feigned disabilities such as blindness" were trying to curry sympathy with the wealthy was a common trope (Chess 2003, p. 107). In fact, Katherine Schaap Williams argues that the examination of Simpcox's bodily difference serves to "disclose a character's internal state without his volition" (Schaap Williams 2021, p. 166), taking away Simpcox's

agency to define his own subject position and placing it instead upon Gloucester. Indeed, it is the doubting Gloucester who saves the king from being deceived in this exchange. Michael Harrawood makes an important point about Gloucester's methodology:

> Gloucester—in his last moment of public success before his fall—gets him to identify the colors red and black. Since he is claiming to be blind from birth, the words "red" and "black" must necessarily have no connection for him with things in the world (like the men's cloaks); but Simpcox has made an even bigger mistake by using similes to identify the colors: "red as blood" (2.1.107) and "coal-black as jet" (2.1.110). Simpcox gives himself away because his common sense could not possibly summon these correspondences without the connection in the wits of the referent colors of red and black. (Harrawood 2007, p. 89)

Harrawood's point about "common sense" is an important one, because the notion of common sense inflected both gendered and religious prejudices in the period that would have likely resonated with Shakespeare's audience. Simone Chess points out that Gloucester's interrogation of Simpcox "stage[s] the scientific process through which this deception was uncovered" (p. 107). The scientific process in this instance is counterintuitive to Henry's faithful acceptance of the miraculous, and is tied to reason and temperance in the period, qualities associated with maturity and masculinity. As Alexandra Shepard explains, "Conduct writers... equated manhood with reason, temperance, and self-control and labelled deviation from these virtues in antithetical terms of unmanliness, beastliness, or effeminacy" (Shepard 2008, pp. 29–30). Henry's inability to participate in that form of interrogation highlights his childishness, rather than his adulthood. Henry's ingenuousness situates him clearly within the realm of childlike behavior that was, at the time, a common fault of young men whose manliness could be suborned if not properly cultivated (ibid., p. 28). We know that Henry was a boy king, so perhaps his naivety is appropriate for a young man, but, at the same time, he rejects the lessons in skepticism offered to him by his retainers.

Therefore, Henry's credulousness highlights his inexperience, an inexperience that spells trouble for his subjects. In the case of Simpcox, Henry takes Simpcox's piety for granted, while overlooking a commonly-held prejudice that the poor were trying to deceive their social superiors. Lindsey Row-Heyveld explains that "the involuntary poor became suspect, since, by the action of begging, they signaled dissatisfaction with their station in life, a condition regarded as sinful and rebellious" (Row-Heyveld 2009, p. 4). In Shakespeare's day, acting without suspicion towards the poor would have made Henry look foolish. Gloucester's suspicions of Simpcox highlight the idea that Henry's faith leaves him exposed to potential threats. Indeed, Henry's inability to smell the threats in his own court further accentuates this problem. The same people who are supposed to be helping him learn to approach the world with a healthy dose of cynicism are also untrustworthy; in a sense, Henry is right to be slow to listen to them, even though some of their lessons in skepticism could potentially protect him from their deceit. In the play, Henry chooses between York and Somerset to be the regent in France, but both are bad choices. York stresses his worth (2.1.3.104-104) as a servant, despite the fact that he has already been plotting against the king, while Suffolk was thought to have sold entire towns back to the king's enemies (2.1.3.236-138). Had Henry practiced the kind of skepticism modeled by his own advisers, perhaps he could have avoided the conflict that arises out of this choice altogether.

There is another layer to Simpcox's hoax that bears observation in this context, as well, one that is specific to Simpcox's fake blindness. Simpcox is impersonating someone who is blind so that he can ascribe a false miracle to Henry's arrival. In a sense, Shakespeare relies on the disability discourse of the period to provide a new twist on the morality plays of the past. Blindness was not universally treated as an impairment. Because the eyes provide access to the physical world, rife with temptations and danger, people who were blind were sometimes seen as more pious, or at least less inundated with sin. While it is often true that moral judgement was cast upon those who experienced disability, especially in the case of

Richard III, some impairments might actually prevent someone from engaging in sinful thoughts and deeds: "Sightedness, for example, could restrict insight" (Iyengar 2015, p. 5). Schaap Williams has described how Falstaff uses his bodily difference to offer "an etiology for his disabling that works to his advantage" (Schaap Williams 2021, p. 164). I have argued elsewhere that a similar process happens in *Richard III*, where Richard relies on his enemies judging him for his bodily difference to empower himself (Carter 2021, p. 35). Simpcox's pseudo-disability works in a comparable way, in that he tries to direct the king's reading of his newfound sight toward evidence of his piety. Interestingly, though, Simpcox might be seen to not only feign a disability in order to "miraculously" recover from it—he might also be accidentally feigning a kind of moral exceptionalism that he has now lost by "recovering" his sight. Despite the fact that Simpcox was never blind in the first place, his change in status from "blind" to "sighted" might also be seen as a change in moral status from "enlightened" to "carnal"—the exact opposite effect he hopes to bring about. I think we should be careful of essentializing disability, even through positive stereotyping, but it seems useful to our understanding of the plays to consider the ways in which Simpcox's so-called miracle actually reflects the way doubt and knowledge operate in the trilogy.

Simpcox's eyes are opened in a context that also refutes the legitimacy of miracles (or, at least, the specific miracle he is trying to sell to the king). When Suffolk asks Simpcox how he came to be blind, he answers in a seemingly-contradictory manner:

Suffolk: How cam'st thou so?

Simpcox:                          A fall off of a tree

Wife: A plumb-tree, master.

Gloucester:                  How long hast thou been blind?

Simpcox: O, born so, master.                                                      (2.2.1.93-95)

While Simpcox appears to recover from the error when prodded about it (he amends that "all my life" means "since I was young"), we might put some pressure on this very particular account of events. How, in other words, does falling out of a tree lead to blindness? We might interpret this as a reference to the Fall of Man, the primordial event in Eden when Adam and Eve's disobedience brings sin into the world through plucking fruit from a tree. Without a doubt, Simpcox is not sly enough to answer in these sorts of half-truths. Shakespeare, on the other hand, might well be using the imagery of the Fall to make sense of Simpcox's nonsense. In other words, Simpcox's choice to feign blindness, i.e., to lie, is a product of original sin, and his condition of not-really-blindness being healed represents a transition from innocence and into guilt. Just as Adam and Eve are enlightened to the fact that they are nude in Eden when they doubt God's injunction against taking from the tree, the courtiers are enlightened to Simpcox's impiety. Here, too, we see (and hear) a tension between the need to seek knowledge through skepticism and the aspiration towards faithful naivete. The audience sees Henry discovering and punishing the guilty through canny application of skepticism, but at the cost of losing a miracle, and with it, an opportunity for religious celebration.

To take the metaphor further, the king demonstrates a kind of blind faith, which is contrasted to his courtiers' doubt-fueled sight. Revealing Simpcox to be a charlatan reveals the primacy of sin over piety, and the miracle of sight returning to the blind is thereby supplanted by another miracle, one that fits a different understanding of Christian teachings. Gloucester offers to show a miracle of his own, one "That could restore this cripple to his legs again" (2.2.1.129). Instead of letting Simpcox witness against himself, as he accidentally does in the interrogation regarding his blindness, the courtiers show he can walk by whipping him and chasing him away when he leaps up in pain (2.2.1.145.1-2). What is noteworthy about this event is the fact that Gloucester has been shown, throughout the play, to represent a kindness and nobility, but in this moment, he acts very differently, carrying out a kind of torture as propitiation for Simpcox's lies. The oblique reference to the Fall that Simpcox utters in his interrogation sets this moment up with teleological symmetry—Simpcox's fall from the tree leads to his sin, which is paid with torture.

The king's response to the event is telling. He raises his eyes to Heaven and utters "O God, seest thou this, and bearest so long" (2.2.1.146). The Queen replies, "It made me laugh to see the villain run" (2.2.1.147). The court, exemplified by the queen, sees the subversion of the miracle as a source of entertainment, even joy, while the king's response to this enlightenment is to long for God's justice over sin. In the larger scope of the trilogy, then, we see the king's eyes consistently turned away from the world, which makes him credulous, but his subjects are doubtful. Their doubt exposes the sinfulness of the king's subjects, a guilt that he seems unable to see for himself. In that way, King Henry's faithful piety characterizes him as a Christlike figure, while his courtiers' incredulity may ground them in reality, but it ultimately also means that they doubt his leadership, too. The King's upturned eyes make him blind to the schemers and threats that surround him, while his doubtful subjects' trenchant vision cause them to focus on those same threats to the detriment of social cohesion. Their doubt in his rule may well be justified; Henry is not a good ruler. However, the materialism of the court seems to bring Henry down to earth in ways that undermine the miraculous.

If Simpcox provides a perverse metaphor for the Fall in *2 Henry VI*, Joan of Arc provides a parallel metaphor in *1 Henry VI.* Joan is obviously Shakespeare's most egregious example of character assassination in the trilogy, and scholars have naturally gravitated to the way that she is equated with violent femininity and witchcraft. Part of the challenge Shakespeare doubtlessly faced in representing Joan to his Protestant audience was Joan's affiliation with Roman Catholicism (while Joan of Arc would not be canonized by the RCC until the 20th Century, her status in France as a pious folk hero was well-established by Shakespeare's time). Joan was a figure who inspired faith in the French people, and, as the story has been told ever since it took place, her pious access to God inspired her to turn the tide of the wars in France's favor. Patrick Ryan, however, points out that Shakespeare obliquely references apocalyptic imagery associated with Joan by chroniclers such as Edward Hall to suggest that she empowers herself through congress with the devil. However, as Ryan astutely points out, "Shakespeare, despite the Tudor chroniclers' expressed condemnations, gives no conclusive dramatic evidence through most of *1 Henry 6* that Joan receives power from Satan, not, as she claims, from God" (Ryan 2004, pp. 60–62). Only at the end of the play do the demons appear, and even then, they reject her pleas for help. It is likely, as Ryan suggests, that Shakespeare's audience was uniquely attuned to Apocalyptic imagery at the time the play was performed, with the memory of Mary's martyring of English protestants still freshly preserved in the writings of John Foxe (ibid., p. 63). It is worth considering, however, the ways in which Joan is paralleled to Simpcox, and what Shakespeare might be trying to say by inviting those comparisons, especially when the demons eventually arrive to disown her.

To start with, Joan is interrogated by the French court in much the same way that Simpcox is interrogated by Henry's retainers. At the beginning of the play, Exeter speaks of the French in terms of their misplaced faith:

> Shall we curse the planets of mishap
>
> That plotted thus our glory's overthrow?
>
> Or shall we think the subtle-witted French
>
> Conjurers and sorcerers, that, afraid of him,
>
> By magic verses have contrived his end? (1.1.1.23-26)[3]

The question is telling—the French are both "subtle-witted" and deceived by the devil at the same time. However, we see something of a different story in the French court itself. Almost comedically, the following scene paints a very different picture of the French. The Bastard of Orleans enters the court promising an audience with a true prophet in the form of Joan:

> The Spirit of deep prophecy she hath,
>
> Exceeding the nine sibyls of Rome:

What's past and what's to come she can descry. (1.1.2.55-57)

In other words, the Dauphin is given an opportunity to believe in Joan's spectacular powers, just as Henry is given the opportunity to believe in Simpcox. Noteworthy, however, is the fact that the Dauphin declines the invitation to believe, instead putting Joan to the test, just as Simpcox experiences among Henry's courtiers. The Dauphin changes places with Reignier in order to "sound what skill she hath" (1.1.2.63). Taken together, the two opposing courts have the same approach to skepticism, with one important distinction: in the English court, it is the courtiers who are skeptical, while the king is quick to place his faith in spectacle. Meanwhile, in the French court, the Dauphin is hesitant to believe in Joan, and demands that she show her abilities through a lavish display. Because Charles demands a display before he is willing to place his faith in Joan, his faith is conditional upon proof, rather than supplying proof where none exists. Whereas Henry is faced with evidence that consistently undermines his faith, the Dauphin tries to shore up his faith with evidence.

In the case of the Dauphin, however, Joan is able to supply the spectacle he requires to believe in her abilities. While Joan is immediately able to spot Reignier's deception and find Charles in the crowd, despite having never met him, this trick itself is not proof enough for the Dauphin. To prove the validity of her claims, Charles demands "In single combat thou shalt buckle with me" (1.1.2.95), declaring "thy words are true" (96) if Joan is able to win. Her quick victory leads him to declare her dominance in notably religious terms: "Thou art an Amazon/And fight'st with the sword of Deborah" (1.1.2.104-105). Comparing Joan to Deborah, well-known for slaying Canaanites in the name of Israel, is no incidental allusion—his faith in her is declared in religious terms. Charles's response to her is faith-infused, but also contains a hint of anti-religiosity, perhaps even a gesture towards the ongoing disagreements about the evidentiary value of trial-by-combat. When Joan assures him that the Virgin Mary helped her overcome him, he responds:

Who'er helps thee, 'tis thou that must help me.

Impatiently I burn with thy desire. . .

Let me thy servant and not sovereign be.

'Tis the French Dolphin sueth to thee thus. (1.1.2.107-8, 111-112)

The combination of religious and sexual imagery seems, perhaps, heretical, but I would argue that this phrasing offers one of two ways forward; Charles can be a conqueror in a religiously-tinged war, or he can be a conqueror in love. Joan, however, directs his energies to the former. The scene establishes Joan's authority as a religious figure and as the savior of France, though it could be argued that her defeat of Charles serves to effeminize him, rather than authorizing Joan. As the play continues, however, we can see that Joan's victory over Charles is not simply a condemnation of Charles as a weak king. Her interactions with him serve as yet another vehicle by which Shakespeare complexly navigates the themes of belief and doubt, holding up another example of a naïve king to whom Henry can be compared.

Only Talbot is able to mount a meaningful defense against Joan's wiles, and it seems not-incidental that this comes from his recognition of her as a real spiritual power, but not possessed of a power that comes from God, a position he repeats throughout the play:

Foul fiend of France, and hag of all despite,

Encompassed with thy lusty paramours. . . (1.3.2.51-52)

Talbot imagines his one-on-one fight with Joan in terms of a holy war against the forces of evil. While Charles's loss to Joan serves as an aphrodisiac, when Talbot meets her on the field, he conceptualizes his behavior as explicitly religious. "Devil or devil's dam", he intones, "I'll conjure thee. . . And straightway give thy soul to him thou serv'st" (1.1.5.5, 7). When he is initially bested by her, he regroups and wrangles with Joan again: "Heavens, can you suffer hell so to prevail" (1.1.5.9). Talbot's ultimate victory over Joan positions him as a hero in a spiritual war of sorts, one that is noteworthy for the way doubt and

faith animate each army. The moment echoes the way that Protestants criticized Catholic ritual in the period—iconoclasm and metaphor were valued over sumptuosity and ritual. The French rally around their misplaced faith in Joan, and the mystical spectacles that she enacts, and have at least temporary success in the war as a result of that faith, while Talbot's faith has no spectacle around which to rally; he simply states that Joan is a witch and goes about the business of fighting against her. While the battles themselves may be spectacular, Talbot displays no mystical signs, and he needs visual proof of Joan's alleged witchcraft; he acts directly on faith in his own righteousness. As the courtiers in *2 Henry VI* exercise doubt as a means of sheltering Henry from deception, Talbot doubts that French faith is properly placed, and exercises his own faith as a counteragent to the demonic influence to which they succumb. As John A. Warrick states, "Shakespeare draws upon the incarnation to emphasize the willingness of human characters to manipulate Christological associations" (Warrick 2016, p. 65). While Warrick is specifically referring to the play's evocation of the Harrowing of Hell, I argue that the figure of Talbot's faith in things not seen (in this case, the righteousness of his cause and the religious authority under which he defends that cause) is both a manipulation of Warrick's "Christological associations" and a reflection of how the trilogy juxtaposes faith and doubt for dramatic effect.

Despite mixed results in her bouts with Talbot, Joan consistently overwhelms the men she encounters, and the context is always brewed in a combination of martial and gendered threats. When she turns Burgundy, for instance, he exclaims:

> Either she hath bewitched me with her words,
>
> Or nature makes me suddenly relent...
>
> I am vanquished: these haughty words of hers
>
> Have battered me like roaring cannon-shot. (1.3.3.58-59, 78-79)

The idea that Joan's powers are either bewitchingly or militarily overwhelming have a similar effect—Joan's power allows her to overwhelm the defenses of the men she encounters and cow them; it takes a hyper-masculine resistance, like that of Talbot, to stand a chance against her.

The reference to witchcraft in this scene is, of course, not an accident. As we learn in Act V, Joan has been communing with demons, and only at the end of the play do they ultimately fail her in her time of need. She summons them:

> This speedy and quick appearance offers proof
>
> Of your accustomed diligence to me.
>
> Now, ye familiar spirits, that are culled
>
> Out of the powerful regions under earth,
>
> Help me this once, that France may get the field. (1.5.2.29-33)

Despite her entreaties, the demons fail her, and England ultimately takes the field. While evil forces do not come to her aid now, her admission that they have had "accustomed diligence" makes it clear that she has been communing with the devil, and the fact that she has fed "you with my blood" (35) makes it clear that they are her familiars, and that she is, in fact, a witch. The difference between Joan and Simpcox, then, is that one is faking a miraculous spectacle, while the other one is actually delivering it, albeit using forbidden powers to do so. If Henry is condemned for being too naïve about Simpcox's cured blindness, the French court is condemned for placing faith in real powers that are undeserving of their fealty. Joan's witchcraft is thereby represented not as a false miracle, but a miracle enacted by vile means, coded with the same level of judgment that Protestant polemicists levied against Catholic ritual.

Of course, witchcraft is not limited to Joan; in fact, it pervades the three plays. For instance, Eleanor, the Duchess of Gloucester, consorts with the witch Margery Jourdain, and happens to similarly highlight the disconnect between belief and knowledge Shakespeare manipulates. According to Nina S. Levine, because Eleanor is both aggressive and a woman, early modern audiences would have easy access to tropes that characterized her as a witch

([Levine 1994](), pp. 104–5). Further, Stephanie Holden argues that Elanor is characterized as Catholic in juxtaposition to Henry's ahistorical Protestantism ([Holden 2022](), p. 136). Because Eleanor represents a kind of Catholic superstition, and because she is a woman who dabbles in witchcraft, she parallels the way Shakespeare positions Joan as a threat to English Protestantism. Margery Jourdain being friends with Eleanor likewise creates dramatic problems for Shakespeare. As Holden argues, their friendship "suggest[s] Shakespeare found the possibility of women in solidarity to be too great a threat to represent in a play wherein peasants were already rebelling" (ibid., p. 139). Indeed, Jourdain barely speaks in the play, and only enters so that she can summon the demon that foretells the violence that is yet to come. Yet, silent though Jourdain is, her power is clear and absolute. The spectacle of her demonic summoning would be a sight to see onstage, but the truly impressive point is that the demon correctly foretells the future. Unlike Simpcox's spectacular healed eyes, and unlike Joan's communications with Mary, Jourdain's powers are demonstrably real in the play. The demon actually comes, follows her orders, and speaks the truth. Stephanie Holden points out that the punishment Elanor and Margery endure when they have been caught is surprisingly lenient. Elanor's exile, while humiliating, is not death, and Margery is not burned at the stake, as would be expected (ibid., pp. 139–40). She points out that one reason for this may be that "Eleanor's motivations are interpreted as manufactured by and for men" (ibid., p. 139). We certainly should not discount the very real gendered associations between powerful women and witchcraft, which the play does plenty to accentuate—especially when we consider *Richard III*, in which prophesies are relegated to men, while curses are the miracles performed by women, causing the gendered associations of witchcraft to become even more glaring. Richard fabricates prophecies to mislead his opponents (1.1.39),[4] while he himself remembers prophecies set out by Henry himself, which do come true (4.2.95). Meanwhile, Margaret's curses come true, but only in portentous ways (3.3.14).

There is an important extra layer, however, when viewing these events through the lens of doubt/faith. Aside from Henry, the faithful characters in Shakespeare's trilogy mainly place that faith in demonic forces (as with Joan). While scholarship often suggests that the play posits a skepticism of religion itself, I would argue that what we see in *1, 2,* and *3 Henry VI* is an important benefit of skepticism. It grounds faith and makes it stronger. While Henry may seem naïve and the witch-followers may seem demonic, the skeptics of the English court, who show little evidence of faith, likewise tear down the social order and bring disruption and ruin upon Henry's campaigns. That is, except for one particular figure: the Duke of Gloucester.

It is, perhaps, telling that Shakespeare positions Elanor as a witch, given that her husband so surprisingly helps unmask and torture Simpcox later in the play. Eleanor, like Joan, attests that she can foretell the future, but when she tries to convince Humphrey to overthrow Henry and claim the throne for himself—"Henry and Dame Margaret kneeled to me,/And on my head did set the diadem" (2.1.2.39-40)—Gloucester is not fooled by the prophecy. Instead, he recognizes it as a plot on Eleanor's part, not a genuine prognostication, stating:

> And wilt thou still be hammering treachery
>
> To tumble down thy husband and thyself
>
> From top of honour to disgrace's feet?
>
> Away from me and let me hear no more! (2.1.2.47-50)

Gloucester shows the same skepticism that he will later show toward Simpcox in this moment, and immediately recognizes her actions as treacherous and devoted to "worldly pleasure" (2.1.2.40). This is yet another example of Shakespeare pitting faith against knowledge, but, in this case, the frequently canny Humphrey knows that his wife is untrustworthy and is saved from accidentally turning the play into *Macbeth* as a direct result of his critical thinking skills.

What may be the most interesting element of the knowledge/faith dichotomy in this play is the fact that Humphrey expresses a kind of skepticism that would be legitimately helpful to the English monarchy. Just as he helps save Henry from Simpcox's misbehavior, he also expresses skepticism of the traitors who undermine Henry's rule. However, the traitors who overthrow Henry are also aligned with a type of skepticism and cynicism that makes it hard to draw clear lines around who is and who is not undermining Henry's efforts. Essentially, Henry has a complete, blind faith in the goodness of humanity that the play completely rejects. Gloucester has skepticism of everyone and everything except for Henry, and while he does stand up to some of Henry's ill-advised decisions, he still supports the king to his own detriment. Meanwhile, the rest of the court seems to occupy a position of complete skepticism, especially about Henry's reign, which ultimately leads them to failure. If Humphrey keeps a balance between faith and doubt, his fall from grace only serves to drive home the point that Henry's particular government is polarized by the Wars of the Roses, and that the extremes of faith and doubt which characterize the two factions only lead to ruin. Everyone turns their backs on Gloucester because his balance of faith and doubt is unsustainable for those who live in the extremes of each. Like the French in *1 Henry VI*, whose skepticism leads them to be follow Shakespeare's witchy Joan to complete ruin because of their misplaced faith, the English spend more time undermining Henry than offering him good counsel and support.

Perhaps no other segment of the trilogy highlights the disjointed nature between knowledge and faith more directly than the duel in *2 Henry VI*. While this duel is not religious in nature, as the previous examples I have explored here are, the purpose of judicial combat carefully treads the same line between knowledge and faith as the other examples. The purpose of judicial combat is to determine who is being honest and who is lying. The conceit behind it was that God would not allow a person to die in a fight if they were defending the truth. In fact, as Jennifer Low explains, "Because of the religious and moral implications of the oath, and because of the religious assumptions on which the rite of the duel was based, the rite itself was invested with a ceremony of gravity associated with Church rites" (Low 2002, p. 13). Therefore, while trial by combat was not considered a religious spectacle or miracle in the same way as a person who was blind regaining their sight, it also epitomized both the concepts of knowledge-as-observable-fact and faith. The knowledge of who had lied and who was speaking the truth was hidden knowledge that the duel was intended to provide. Meanwhile, the notion that the duel itself was under the protection and direction of God was entirely faith-based. Even though the practice was allowed to proceed, and even though the premise of allowing this practice to exist was that God would intervene on the side of right, the victor's guilt or innocence was still investigated after the fact (ibid., p. 14). Therefore, a great deal of faith was placed in the outcome of a duel, while skepticism still found its way into an interrogation of the practice after the fact.

So, in *2 Henry VI*, when Horner and Peter are at odds about whether or not Horner had cast doubt on Henry's right to the throne, it is decided that they should duel in order to discover the truth. Yet again, Henry exercises a kind of blind faith by acceding to his lords' wishes that a combat be set up. Peter protests that he "cannot fight" and "shall never be able to/fight a blow" (2.1.3.214, 216-217), but Henry accepts Humphrey's suggestion and allows the fight to proceed anyway. To say the least, it is difficult to imagine a judicial combat is fair when one of the fighters insists he is unable to defend himself, which would in turn cast doubt on the outcome of the duel. Yet Henry is willing to allow the fight to proceed, as well as to replace York with Somerset in France, simply because the possibility of Horner's guilt casts doubt upon York (as Horner's master). When they actually fight, Horner is beaten down by the supposedly-weaker Peter, but not before he confesses to his wrongs: "Hold, Peter, hold! I confess! I confess treason" (2.2.3.96). One could interpret the outcome of the duel as a kind of miracle in itself—if Peter truly believes he is unable to fight, yet overcomes the well-trained Horner, who we learn "hath learnt so/much fence already" (2.2.3.79-80), then his victory is like the sort of miracles we have seen undercut

time and time again throughout the trilogy. However, because Horner confesses his treason before his death, Henry and the rest accept the outcome of the duel, with Henry promising Peter a "reward" (2.2.3.107). Faith in the duel's veracity is rewarded with confirmation, or knowledge, that Horner had, in fact, spoken the treasonous words, despite the doubtful circumstances of the duel itself. In this moment, the tension between faith and doubt that Shakespeare has been toying with for a play-and-a-half coalesces into a moment of clarity. The duel-as-spectacle serves the turn of both knowledge and faith, providing knowledge *as a result of* placing faith in the outcome of the duel itself. The pain and failure experienced by characters who are too naïve is balanced with the skepticism that leads to testing that faith (in this case, through violent acts). The duel perfectly balances the two ideologies, despite their seeming incompatibility. Shakespeare seems to support a praxis of faith, but not without a healthy dose of skepticism and experimentation. The duel serves as a model for how belief and knowledge can support one another, rather than existing separately from one another, as it is described in Hebrews.

Throughout the trilogy, characters fail to find a balance between faith and doubt. In *3 Henry VI*, Lady Grey fears that Warwick is coming to depose Edward and reclaim the throne for King Henry. She makes a notable statement: "For trust not him that once hath broken faith" (3.3.4.30).[5] The line is dripping with irony, though Lady Grey may not realize it. After three plays of broken pledges, betrayals, and reversals, she now defaults to measuring a man based on whether or not his word can be believed. There is something almost sad about the declamation—Edward would never have been king in the first place if not for dishonesty, and yet his wife upholds honesty as a superior masculine virtue. Alexandra Shepard explains that, in the period, honesty was thought to have been bestowed upon married men by their wives when they wed (Shepard 2008, p. 64). Naturally, faith is a requirement of trust—if one *knows* something is true, there is no need for trust. When Lady Grey besmirches Warwick's honesty, she does so because she privileges faith over skepticism. And yet, as we have seen throughout the three plays, poorly balancing faith with skepticism leads to disorder.

Shakespeare's *Henry VI* trilogy is unusual among his writings because of its emphasis on religion, religious thinking, and the accompanying spectacle. From Henry's speed to accept the legitimacy of Simpcox's healed eyes to the French Dauphin's misplaced faith in Joan, Shakespeare negotiates the idea of faith in the plays in a complex way. On one hand, faith is laudable and aspirational, but faith with a healthy dose of skepticism, while oxymoronic, seems to be a better recipe for success. In a discussion about the disconnect between natural laws and spiritual ones, Maurice Hunt points out that:

> In this early history play [*3 Henry VI*], Shakespeare boldly interrogates Christianity's solution to the deadly laws of nature by stressing how unnatural and equally self-destructive it can be for certain individuals in certain circumstances. Represented or inferred Christian values. . . often appear attractive but only when they are considered as ideals abstracted from their local dramatic contexts of presentation. (Hunt 2008, p. 155)

For Hunt, the tension between practiced Christianity and pragmatism is one that makes the *Henry VI* plays stand out on a thematic level. So, too, with the tension between faith and skepticism, Shakespeare challenges easy answers about the place of faith in his society. Moments of knowledge acquisition are often suspect, such as Simpcox's failed hoax, while moments of blind faith can also be misleading (Henry's acceptance of Simpcox's story is only possible because of his naivete). Therefore, we must consider the effect of each of these practices, rather than only their aspirations. Practicing skepticism may seem noble when Gloucester and the others save Henry from being deceived by Simpcox, but their attitude toward his faithfulness ultimately undermines his power and leads to their own squabbling. Meanwhile, Henry shares with the French court a predisposition toward faithfulness, but that faith is often misplaced, empowering witches and charlatans. Indeed, as in Hebrews 11:1, Henry uses faith to fill in the gaps of his knowledge, but does so to such an extent that he ignores knowledge that *is* attainable in lieu of letting faith take the

lead. Taken as a whole, Shakespeare seems interested in the ways that faith is necessary for social cohesion, while refusing to outright condemn those who practice healthy skepticism. Through negative examples of characters exhibiting too much faith or too much skepticism, Shakespeare illuminates a third way, in which skepticism can build faith, while faith leads to otherwise unattainable knowledge. Henry's failures as a king often stem from his refusal to interrogate the world around him, while his court lets him down by failing to offer good counsel and loyalty. As a result, England tears itself apart, loses France, and makes space for Richard III to take the throne.

**Funding:** This research received no external funding.

**Informed Consent Statement:** Not applicable.

**Conflicts of Interest:** The author declares no conflict of interest.

## Notes

[1] In this essay, I shall discuss the three *Henry VI* plays as Shakespeare's for the purposes of simplicity. However, recent scholarly consensus has made it clear enough that Marlowe and others had a hand in the plays' composition to such an extent that *The New Oxford Shakespeare* lists the authors as "William Shakespeare, Christopher Marlowe, and others" including "Thomas Nashe" and "Anonymous" as the playwrights (Taylor et al. 2017, pp. 255, 335, 927). Therefore, ascribing representations of characters such as Joan of Arc to Shakespeare is more in service of simplicity than an attempt to ascribe to Shakespeare what may have been the work of others. Similarly, it is well-acknowledged that *1 Henry VI* was written out of sequence, and so readers should bear this in mind as I discuss the plays' connections to one another.

[2] All citations of this text come from The Arden Shakespeare: King Henry VI Part 2 (Shakespeare 1999).

[3] All citations of this text come from The Arden Shakespeare: King Henry VI Part 1 (Shakespeare 2000).

[4] All citations of this text come from The Arden Shakespeare: King Richard III (Shakespeare 2009).

[5] All citations of this text come from The Arden Shakespeare: King Henry VI Part 3 (Shakespeare 2001).

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
