# Peer review of "“O Piteous Spectacle! O Bloody Times!”: The Faithlessness of English Identity in 1, 2, and 3 Henry VI"

_religions, doi:10.3390/rel15010013_

Round 1
Reviewer 1 Report
Comments and Suggestions for Authors
The essay’s central argument is clearly defined and contributes to an ongoing and vibrant conversation on the place of “belief” in Shakespeare’s plays. Here, the author is interested in how the Henriad negotiates a healthy “skepticism” by pointing to Henry’s too easy religious faith in contrast to those of the courtiers around him, some of whom are trying to take advantage of that disposition for their own gain, and some of whom are looking to protect the king and his kingdom.
There is some merit to what the paper is exploring and it nicely thinks through the political tensions in those plays. While it does address the issue very well in 1, 2 Henry VI and at some length, the paper comparatively engages little with 3 Henry VI, except at the very end. I wonder if that section could be interrogated more, or if it is the case that the writer sees that there is less to support as strong a claim that governs that play?
I know the article limits itself to the Henry VI plays, but is there room for Henry V along these same lines of inquiry? That is, it would be fascinating to consider if the older Shakespeare in 1599, revisiting the cycle and its tensions in the “prequel,” strikes some kind of balance between faith and skepticism in his portrayal of Henry V, a king whose faith becomes in one sense the center for the justice of his cause in France that is then “proved” to be “true” in his unlikely defeat of the French armies. Attending to H5 would help fill out the writer’s treatment of the tetralogy.
A few questions:
1. The writer assumes that H6’s religious faith is one of the causes of collapse, but is correlation the same as causation as depicted in the plot?
2. Can “faith”—and its liabilities—be expanded to include not just religious faith, but faith more widely conceived in other “categories,” including the loyalty of the feudal bond or chivalric action? How do the plays put those other “faiths” to the test?
A few minor things:
1. The writer claims that the Henry VI trilogy is often pointed to by critics for its “more-than-usual religiosity” by critics, but the writer cites only one source (J-C Mayer) to situate the argument.
2. At one point, the writer gestures towards Shakespeare’s own “skepticism of trial-by-combat” (ll. 273-274). That may be the case, but I’d stay away from attributing anything to Shakespeare himself; perhaps if the writer can lodge the claim within a contemporary cultural trend (dueling, for instance, which Elizabeth outlawed in 1571 to no avail) to which Shakespeare might be responding?
3. The authorship of Hebrews is contested (l. 55), and there is no scholarly consensus that Paul is the author.
4. On line 108, the writer asserts that the “scientific process” is “coded as inherently masculine.” Can this point be clarified?
Author Response
This feedback was excellent and very helpful.
I agree that the section on 3 Henry VI is shorter than the other two, and as the reviewer suspects, I think there's less to say about that play than the other two.
I think it's a great idea to continue this interrogation of Henry V, and may write a second article on 1 HIV, 2 HIV, and Henry V at some point in the future. For the purposes of this article, I worry that adding that would make my overall points too diluted without doing justice to the excellent suggestion.
A few questions:
- I think this is a great question. I suspect the play is blaming someone, whether it's Henry for his naivete or his courtiers for their lack of faith in him. I think the causation is baled into the way they mismanage the war.
- I have tried to gesture toward this within the revision. I don't think I can offer a thorough examination of that without significant increase to the length of the analysis, but I make the point that fealty, etc. are synecdochally related to religious praxis.
A few minor things:
- I included a second reference to make this more-thorough without going for a full literary review.
- I think my prose was unclear here, and I've revised to make this less fuzzy.
- I didn't know this - I've updated, with thanks!
- Again, I think I was unclear here, so I updated accordingly.
Reviewer 2 Report
Comments and Suggestions for Authors
This is a well-researched, thought-provoking, and engagingly written and argued exploration of religious belief and its relationship to doubt in the three parts of Henry VI. The use of Hebrews 11: 1 (‘faith is the grounds of things which are hoped for, and evidence of things which are not seen’) provides a really helpful framework for thinking about the plays’ treatment of the supernatural - or the supposed supernatural. The examples which are discussed are well chosen, and without exception the close readings tease out the complexities, ironies, and paradoxes of the stage action. The article is also alert to the broader narrative patterns and structural parallels that unfold within individual plays, and across the three plays as a whole
The article is joining a very crowded critical field. There is a large and ever-expanding critical literature dedicated to the first tetralogy, and there is an even larger body of work concerned with Shakespeare’s engagement with religion and the supernatural, which itself encompasses the place of doubt and scepticism in his world view. The author does a first-class job of setting out their place within this critical domain, and indicating the contribution the article is making to our understanding of the plays. However, I do think the author - even if only very briefly - should take some account of the issue of the authorship of 1 Henry VI, and the sequence of composition of the three plays, which itself prompts the question of whether the plays form a continuous, overarching whole. I recognise that these issues are tangential to the focus of the article, so I’m not suggesting that the author engages substantively in these arguments. But I do think it is important briefly to acknowledge that the scholarly consensus is that 1 Henry VI is a work of collaborative authorship, and that there is a strong likelihood that it was written after the composition of the second and third parts. As I say, I’m not looking for a detailed engagement with these textual questions, rather an acknowledgement of them - perhaps in no more than a paragraph - and a recognition that this necessarily complicates the discussion of Joan, in particular. (Can she be so confidently attributed to Shakespeare as the author currently implies?)
Author Response
This is amazing feedback, and I think the article is stronger after considering these generous comments.
I'm glad the Hebrews verse provided useful context. It was a point of some discussion with my editors about how to make it clear and useful.
I also appreciate the textual concerns raised by the reviewer, regarding authorship and timing of the plays. I wasn't able to figure out how to work it into the prose in a clear way that flowed well, so I included an extensive footnote to acknowledge these critical realities and better-situate my claims.